# Cooperativity between the Ribosome-Associated Chaperone Ssb/RAC and the Ubiquitin Ligase Ltn1 in Ubiquitination of Nascent Polypeptides

**DOI:** 10.3390/ijms21186815

**Published:** 2020-09-17

**Authors:** Arnab Ghosh, Natalia Shcherbik

**Affiliations:** Department for Cell Biology and Neuroscience, Rowan University School of Osteopathic Medicine, 2 Medical Center Drive, Stratford, NJ 08084, USA; ghosha@rowan.edu

**Keywords:** rRNA, r-protein, ribosome, ribosome-associated protein quality control (RQC), ribosome-bound nascent chains (RNCs), ribosome-associated chaperones, Ssb/RAC triad, ubiquitin, ubiquitination of polypeptides bound to a ribosome, ubiquitin ligase Ltn1

## Abstract

Eukaryotic cells have evolved multiple mechanisms to detect and eliminate aberrant polypeptides. Co-translational protein surveillance systems play an important role in these mechanisms. These systems include ribosome-associated protein quality control (RQC) that detects aberrant nascent chains stalled on ribosomes and promotes their ubiquitination and degradation by the proteasome, and ribosome-associated chaperone Ssb/RAC, which ensures correct nascent chain folding. Despite the known function of RQC and Ssb/ribosome-associated complex (RAC) in monitoring the quality of newly generated polypeptides, whether they cooperate during initial stages of protein synthesis remains unexplored. Here, we provide evidence that Ssb/RAC and the ubiquitin ligase Ltn1, the major component of RQC, display genetic and functional cooperativity. Overexpression of Ltn1 rescues growth suppression of the yeast strain-bearing deletions of *SSB* genes during proteotoxic stress. Moreover, Ssb/RAC promotes Ltn1-dependent ubiquitination of nascent chains associated with 80S ribosomal particles but not with translating ribosomes. Consistent with this finding, quantitative western blot analysis revealed lower levels of Ltn1 associated with 80S ribosomes and with free 60S ribosomal subunits in the absence of Ssb/RAC. We propose a mechanism in which Ssb/RAC facilitates recruitment of Ltn1 to ribosomes, likely by detecting aberrations in nascent chains and leading to their ubiquitination and degradation.

## 1. Introduction

Synthesis and maintenance of the functional cellular proteome are critical parts of protein homeostasis (proteostasis). Although the translation machinery has inherent mechanisms to correct translational errors, numerous protein quality control systems maintain proteostasis via the ubiquitin (Ub) proteasome system and a variety of molecular chaperones that function co- and post-translationally [1,2]. One such mechanism, the ribosome-associated protein quality control (RQC), begins to operate as early as during translation. RQC detects aberrant ribosome-bound nascent chains (RNCs) and promotes their ubiquitination, extraction from the ribosomal exit tunnel, and presentation to the proteasome for degradation [3]. To accomplish this task, RQC engages multiple enzymes and protein factors (Figure 1A and [4,5,6]), including the release-like factors Dom34-Hbs1/Pelota-Hbs1L (yeast nomenclature given first; human nomenclature after the slash), which, in concert with Rli1/ABCE1, splits stalled ribosomes into subunits without peptidyl-tRNA hydrolysis, generating 60S•peptidyl-tRNA [7,8,9]. The RING family Ub ligase Ltn1/Listerin ubiquitinates peptides within 60S•peptidyl-tRNA [4], while Rqc1/TCF25 promotes formation of K48-assembled poly-Ub chains [10]. Rqc2/NEMF modifies RNCs with CAT-tails, repeats of alanine and threonine attached to the C-terminus of the recipient peptide in a template-free manner [11]. By cleaving off the tRNA within 60S•peptidyl-tRNA, Vms1/ANKZF1 nuclease liberates peptides from the bulky tRNA molecule [10,12], while the Ub-selective chaperone Cdc48^Npl4-Ufd1^/p97^NPL4-UFD1^ extracts the poly-Ub-modified, tRNA-free peptide from the exit tunnel of 60S for subsequent proteasomal degradation [13]. A member of the AAA+ family of molecular chaperones [14], Cdc48/p97 forms hexamer and upon ATP hydrolysis generates a large amount of energy, underlying its activity in energy-dependent processes [15]. The Ub-binding selectivity of Cdc48 is provided by Ub-binding co-factors, such as Npl4/NPL4 and Ufd1/UFD1 [16,17].

Numerous genetic, biochemical, and structural studies conducted in mammalian and yeast cells indicate an important role of RQC. Dysfunction of RQC results in protein aggregation (partly via interactions among CAT-tails) leading to sequestration of cellular chaperones towards aggregates, causing proteotoxic stress that may manifest in various diseases, including neurodegenerative and cardiovascular disorders, cancer, and aging [18,19,20]. Therefore, it is very important to understand all possible nuances of RQC functioning in the eukaryotic cell, including potential deviations from the conventional RQC mechanism described above (Figure 1A), and to identify new factors that might contribute to quality control of nascent protein chains.

As with RQC, ribosome-associated chaperones such as Ssb/ribosome-associated complex (RAC) and NAC operate on nascent protein chains and represent alternative ways to maintain cellular proteostasis [21,22]. However, unlike RQC, these chaperones assist in precise and timely initial folding of nascent polypeptides and provide protection against unspecific proteases, premature folding, binding to random cytoplasmic proteins, or erroneous modifications (reviewed in [21,23,24]).

The Ssb/RAC complex consists of Ssb1 and Ssb2 chaperones (often called Ssb), which are members of the Hsp70 family with the following domain architecture (Figure 1B, top): the nucleotide binding domain is followed by a flexible poly-linker and the substrate-binding domains [25]. The RAC forms stable heterodimers composed of the Hsp40 J-protein Zuo1/Zuotin and the atypical Hsp70 Ssz1/HspA14, which lacks the C-terminal SBDα ((Figure 1B, top) and [26]). Although the active site of Ssz1 contains ATP-Mg^2+^, the absence of essential catalytic residues makes ATP hydrolysis impossible [26,27]. Structural studies have proposed that Ssz1, anchored to ribosomes by Zuo1 (Figure 1B, bottom), acts as a holding chaperone that directs the growing RNC from the exit tunnel towards RAC-bound Ssb [28].

In Ssb/RAC complexes, Zuo1 is responsible for all interactions with r-proteins and rRNAs from both ribosomal subunits [28,29]. The main contacts with 60S are made through ribosomal helices 59 and 24 of 25S rRNA, expansion segment 27L (ES27L) that serves as a platform for key enzymes responsible for RNC maturation [30], and r-proteins Rpl22 and Rpl31 [26,28,31]. The interaction between 40S and Zuo1 occurs through helix 44 of 18S rRNA, which comprises the decoding center of ribosomes [32]. Structural studies revealed that Zuo1 maintains a dynamic and flexible structure that spans ~190 Å across the ribosomal subunits using a long M domain [28,29,33,34] and Figure 1B. Functionally, RAC acts as a co-chaperone for Ssb by stimulating ATPase hydrolysis via the J-domain of Zuo1 and facilitating interaction with nascent polypeptides [27,35]. Thus, Ssb/RAC forms a tripartite complex bound to ribosomes [36], in which Ssb chaperones RNC formation while RAC acts as a co-chaperone [27].

Ssb/RAC plays an important role in diverse cellular processes and its involvement in early RNC folding during translation is well documented (reviewed in [37]). In fact, Ssb/RAC co-fractionates with translating ribosomes (polysomes) and with 60S and 80S ribosomal subunits [35,38,39], forms physical contacts with multiple sites on ribosomes (reviewed in [25]), and captures the first 20–25 amino acids of RNCs as soon as they emerge from the exit tunnel [40]. Moreover, cells with dysfunctional or mutated Ssb/RAC have increased levels of stop codon readthrough, frameshifting [29], and formation of protein aggregates [35]; they are highly sensitive to misfolding agents [38,41,42,43,44], suggesting an important role for Ssb/RAC as an early stage co-translational protein quality assessor. In addition, numerous biochemical studies supported by the recent selective ribosome profiling (SeRP) data provide strong evidence of cross-talk between Ssb/RAC function and protein translation through the chaperone’s participation in RNC folding [40,45]. This raises the question of whether early-stage Ssb/RAC-mediated folding and early-stage RQC are functionally connected. Although previous studies have implicated Ssb/RAC in translation regulation at the late stages (i.e., during translation termination on poly-A stretches of model substrates, or by facilitating translation of stalling-prone poly-AAG/AAA sequences), as well as during ribosome biogenesis [41,42,46,47], the question of whether this chaperone collaborates with ***initial steps*** of RNC surveillance mechanisms remains elusive. In this study, we provide evidence that Ssb/RAC collaborate with RQC via facilitating accommodation of the Ub ligase Ltn1 on ribosomes, thereby promoting efficient ubiquitination of endogenous nascent polypeptide chains.

## 2. Results

### 2.1. Ubiquitin Ligase Ltn1 and Ribosome-Associated Chaperone Ssb Demonstrate Genetic Interaction

Previous studies have shown that strains defective or deficient in Ssb/RAC activity display inhibited growth under stress conditions [38,44]. To investigate this phenomenon in greater detail, we first generated a yeast strain lacking ribosome-associated Ssb, a major component of Ssb/RAC complex. Since Ssb is encoded by two nearly identical and therefore functionally redundant genes *SSB1* and *SSB2*, we deleted both genes, resulting in generation of the double deletion strain *ssb1∆ ssb2∆* (hereafter, *ssb∆∆*). We then tested the phenotype of *ssb∆∆* in stress conditions using the viability assay. Cells derived from *ssb∆∆* and wild-type control strains were plated on YPD agar plates supplemented with aminoglycoside hygromycin B (HygB) or high concentration of NaCl (0.8 M) and grown at 30 °C, or plated on plain YPD agar plates and incubated at low temperature of 20 °C. In agreement with published data [38,44], we detected strong growth inhibition of *ssb∆∆* but not wild-type cells on plates subjected to high NaCl, HygB or cold stress (Figure 1C). Given that the Ssb/RAC complex acts on the early steps of translation by ensuring correct RNC folding [40,45,48], the inability of *ssb∆∆* to grow under stress conditions might be due to generation and accumulation of aberrant polypeptides, some of which are toxic.

The inhibited growth of the *ssb∆∆* strain under conditions of proteotoxic stress (Figure 1C) therefore could be explained by an overload of misfolded polypeptides beyond the functional capacity of RQC in the absence of Ssb/RAC. In this scenario, increased activity of the ribosome-associated protein surveillance system that eliminates aberrant polypeptides should reduce sensitivity of the *ssb∆∆* strain to protein misfolding agents.

To check this hypothesis, we overexpressed Ltn1 with a C-terminal FLAG tag (Ltn1^FLAG^) using 2 μc plasmid in *ssb∆∆* and in wild-type control cells and examined whether additional copies of the ligase would rescue *ssb∆∆* growth reduction under chemically induced proteotoxic stress conditions. Besides functioning within RQC, the Ub ligase Ltn1 has numerous cellular substrates, as was recently revealed by the proximity-labeling proteomic approach conducted in mammalian cells [49]. This suggests that Ltn1 might represent a limiting factor of the RQC system. We found that overexpressing Ltn1^FLAG^ from the strong *ADH* promoter restored cell growth kinetics in HygB- and NaCl-supplemented plates (Figure 1D), suggesting genetic interaction between Ssb/RAC and Ltn1^FLAG^. To verify these results, we studied the growth characteristics of wild-type and *ssb∆∆* in liquid cultures grown with or without stress (Figure 1E). The observed growth inhibition in the culture derived from *ssb∆∆* strain was primarily due to prolonged lag phase and doubling time, and these parameters were elevated upon HygB or NaCl addition (Figure 1E, compare vector control V for *WT* and *ssb∆∆*). Overexpressing Ltn1^FLAG^ in wild-type cells had no effect on the lag and doubling time (Figure 1E, compare V and Ltn1 for *WT*), while in *ssb∆∆* strain, ectopic expression of Ltn1^FLAG^ significantly decreased these growth parameters (Figure 1E, compare V and Ltn1 for *ssb∆∆*). Together, these results presume genetic interaction between Ssb/RAC and Ltn1, and indicate that Ltn1-dependent RQC and Ssb/RAC function within two alternative pathways of protein surveillance that likely share the same substrate—aberrant RNCs.

### 2.2. Ltn1^FLAG^ Associates with Both 60S and 80S, Promoting Polyubiquitination of Associated RNCs

Next, we conducted a series of experiments to validate expression and functional activity of the generated Ltn1^FLAG^ fusion protein, since the C-terminus of Ltn1 (where the FLAG tag is added) contains the catalytically active RING domain [4]. First, we verified the expression of full-length Ltn1^FLAG^ in wild-type cells via western blotting using anti-FLAG antibodies (Figure 2A). Next, we tested whether Ltn1^FLAG^ expressed in *ltn1∆* cells can rescue the lethal phenotype induced by HygB [4,50]. We detected complete restoration of growth in *ltn1∆* cells transformed with Ltn1^FLAG^-expressing plasmid but not with empty vector control on agar plates supplemented with HygB (Figure 2B), suggesting that Ltn1^FLAG^ is functional. Finally, to validate that Ltn1^FLAG^ is ubiquitination-competent and active during RQC, we used an assay previously developed in our laboratory. This assay is based on separating total cellular yeast lysate into ribosomal species-specific fractions and analyzing ubiquitination status of endogenous RNCs generated in cells [50]. To enrich for Ub-modified polypeptides associated with various ribosomal species, we used yeast strain depleted of Cdc48^Npl4-Ufd1^. In this strain, *CDC48* is placed under the control of a tetracycline-repressible promoter (*P_TET-07_-CDC48*); adding the tetracycline derivative doxycycline (Dox) to the growing yeast culture therefore leads to Cdc48 depletion [51]. *P_TET-07_-CDC48 ltn1∆* cells transformed with Ltn1^FLAG^-expressing construct or empty vector control were grown in the presence of Dox, and cellular lysates were sedimented through sucrose gradients and fractionated. Proteins extracted from individual fractions were analyzed by conventional western blotting using antibodies against Ub, as well as control antibodies against r-protein Rpl3 (to visualize 60S-containing gradient fractions) and anti-FLAG antibodies to detect Ltn1^FLAG^.

Consistent with previously published data [52], we found that placing the FLAG tag at the C-terminus of Ltn1 does not affect catalytic activity of the ligase during RQC, as we detected efficient ubiquitination of RNCs associated with 60S in gradients derived from Ltn1^FLAG^-expressing cells but not from cells transformed with empty vector control (Figure 2C,D and Figure A1, fractions 12–13). Although we detected poly-Ub signal in fractions containing the small ribosomal subunit 40S (Figure 2C and Figure A1B, fractions 8–11), a similar signal was also detectable in the gradient derived from *P_TET-07_-CDC48 ltn1∆* cells transformed with empty vector control (Figure 2D and Figure A1A, fractions 8–11), indicating that the detected poly-Ub products are not generated by Ltn1^FLAG^. Of note, the peak levels of ribosomal species detected by absorbance measurements at 254 nm in the control strain transformed with empty vector are presented in Figure A1A. We also noticed significantly larger peak for the 40S subunit when compared with peak for the 60S subunit (Figure 2C and Figure A1) that was not caused by Cdc48 depletion [50]. Considering that in this experimental setting we maintained yeast culture transformants in the SC medium for a prolonged time, we concluded that minimal nutritional supplements might result in alterations of ribosome biogenesis dynamics, similarly to bacterial cells [53].

Interestingly, this sucrose gradient analysis revealed that Ltn1^FLAG^ also actively ubiquitinates RNCs associated with the 80S monosome (Figure 2C and Figure A1B, fractions 14–16). The distribution of ubiquitinated RNCs among the different ribosomal species coincided with the distribution of ectopically expressed Ltn1^FLAG^, as both 60S and 80S contained detectable Ltn1^FLAG^ (Figure 2C, top panel, fractions 12–13 and 14–16, respectively). Although within the RQC pathway, Ltn1 mostly functions on 60S•RNCs as the E3-ligase substrate [4], presence of ectopically expressed Ltn1 in 80S ribosomal fractions has also been observed in previously published studies [4,6,54].

Thus, besides validating functionality of FLAG-tagged Ltn1, our results demonstrated association of Ltn1^FLAG^ not only with 60S•RNCs but also with 80S•RNCs; moreover, during this interaction, the ligase retained its catalytical activity and ubiquitinated newly generated polypeptides coupled to fully assembled ribosome 80S. For simplicity, we termed this branch of the protein surveillance pathway 80S-RQC, as opposed to the conventional RQC that operates on 60S•RNCs [4,5,6,13].

### 2.3. Ssb/RAC Promotes Ubiquitination of Endogenous RNCs on 80S Species

Our observation that Ltn1^FLAG^ overexpression rescues growth inhibition of *ssb∆∆* cells upon proteotoxic stress (Figure 1C–E) implies that the E3 ligase and the chaperone act within two distinct pathways that functionally overlap during stress by sharing the same substrate. To test whether the substrates are RNCs, we next examined whether association of nascent polypeptides with 60S and/or 80S ribosomal species and their ubiquitination were altered upon deletion of Ssb/RAC-encoding genes.

As before, we used Cdc48-depleted cells to increase amounts of poly-Ub-RNCs associated with 60S and 80S ribosomal species. Consistent with our previous work (Figure 1B in [50]), strong polyubiquitin signal was detected in 60S- and 80S-containing fractions in lysates derived from *P_TET-07_-CDC48* cells grown in the presence of Dox (Figure 3A). Next, we deleted *SSB1* and *SSB2* in the *P_TET-07_-CDC48* background to generate the Ssb-null strain (*P_TET-07_-CDC48 ssb∆∆*), while *SSZ1* or *ZUO1* were deleted to obtain strains with dysfunctional RAC complex activity (*P_TET-07_-CDC48 ssz1∆* and *P_TET-07_-CDC48 zuo1∆*). Cells derived from these strains were grown in the presence of Dox, and cell lysates were analyzed by sucrose gradient centrifugation followed by western blotting as described above.

Strikingly, we detected significant reduction of Ub signal in 80S fractions in the absence of the Ssb/RAC complex (Figure 3B, lanes 4–7), while ubiquitination of 60S-associated polypeptides did not display any significant changes when compared to those derived from the *P_TET-07_-CDC48* parental strain (Figure 3A,B, compare lanes 2 and 6). As expected, deleting *LTN1* in *P_TET-07_-CDC48 ssz1∆* (*P_TET-07_-CDC48 ssz∆ ltn1∆*) completely eliminated accumulation of ubiquitin species associated with 60S (Figure 3C), consistent with the well documented role of Ltn1 ligase in ubiquitinating polypeptides bound to the 60S subunit during canonical RQC pathway [4,6,50]. Taken together, these data demonstrate that lack of Ssb/RAC complex components specifically prevents the accumulation of polyubiquitinated polypeptides associated with 80S ribosomes.

### 2.4. Disrupting the Ssb/RAC Triad Abrogates Ltn1-Dependent Ubiquitination of 80S-Associated RNCs upon Depletion of Cdc48

Although Ltn1 is sufficient for recognition, binding, and ubiquitination of 60S-RQC substrates in yeast and mammalian cell-free systems [5,6,50,55,56], further studies conducted in mammalian cells found that NEMF (Rqc2 in yeast) stabilizes the Listerin•60S interaction and stimulates Listerin-mediated ubiquitination of 60S•RNCs [54]. Thus, the activity of Ltn1 in the crowded cell environment relies on the co-factor Rqc2/NEMF in the canonical 60S-RQC pathway. Analogously, we suspected that Ssb/RAC might function as a co-factor for Ltn1 recruitment to 80S-RQC substrates in yeast; RNCs that fail Ssb/RAC-assisted folding or re-folding may undergo triage by this chaperone and identified as terminally misfolded proteins requiring Ub-dependent degradation. Alternatively, Ssb/RAC bound to 80S ribosome might stimulate Ltn1-dependent ubiquitination of RNCs, possibly by restructuring nascent chains to expose ubiquitination sites for efficient modification.

To distinguish between these two possibilities, we next measured amounts of Ltn1 and Ub signals derived from nascent chains present in 60S- and 80S-containing gradient fractions in cells with or without functional Ssb/RAC complex. Considering that Ssz1 is an essential component of the Ssb/RAC complex [28], we deleted *SSZ1* on the *P_TET-07_-CDC48* background to suppress Ssb/RAC function. The Ub ligase Ltn1^FLAG^ was expressed from a plasmid in *P_TET-07_-CDC48 ssz1∆* and its parental strain *P_TET-07_-CDC48* grown in the presence of Dox. The amounts of the ligase and Ub-modified RNCs co-sedimented with 60S and 80S ribosomal species were visualized (Figure 4A,B) and measured by quantitative western blotting.

We used near-infrared (NIR) secondary antibodies conjugated to infrared dyes to visualize signals that correspond to Ltn1^FLAG^, poly-Ub, and Rpl3. To account for the amounts of ribosomes analyzed and unavoidable experimental variations between gradients, fractionation, SDS-PAGE-based proteins separation, and immunoblotting procedures, we normalized Ltn1 and poly-Ub signals to the signal obtained from control Rpl3 protein present in the same fraction. In addition, for better separation of the 60S and 80S ribosomal species, we used 15–35% sucrose gradients instead of 15–42% sucrose gradients used previously (Figure 2 and Figure 3). For every gradient-derived image, we chose a set of fractions corresponding to 60S and 80S ribosomal species based on gradient-trace peaks and Rpl3 sedimentation (Figure A2) for further analysis. NIR signals derived from Ltn1^FLAG^, poly-Ub, and Rpl3 present in same fraction were digitized and summarized, and the amount of Ltn1^FLAG^ was expressed as Ltn1^FLAG^/Rpl3 ratios (Figure 4C), while levels of ubiquitinated polypeptides were expressed as poly-Ub/Rpl3 ratios (Figure 4D) for 60S and 80S. Consistent with other studies [4,6,54] and our previous experiments using conventional western blotting with HRP-fused secondary antibodies and ECL-based detection (Figure 2), 80S contained significantly lower amounts of Ltn1^FLAG^ and poly-Ub-RNCs than 60S in both strains tested (Figure 4C,D; compare bars from 60S and 80S). Importantly, this analysis revealed that absence of the *SSZ1* gene does not affect levels of either Ltn1^FLAG^ or poly-Ub present in 60S fractions in Cdc48-depleted cells (Figure 4A,B), as the difference between these signals was statistically insignificant (Figure 4C,D; compare 60S bars). Unlike for 60S, levels of Ltn1^FLAG^ co-sedimented with 80S and Ltn1^FLAG^-ubiquitinated polypeptides were significantly lower in *P_TET-07_-CDC48 ssz1∆* cells. These results support the hypothesis that Ssb/RAC facilitates the recruitment of Ltn1 to 80S ribosomes.

### 2.5. Loss of Ssb/RAC Triad Integrity Decreases Ltn1 Association with Both 60S and 80S in Wild-Type Cells

Since ubiquitination and release of modified nascent peptides during RQC is a dynamic process, we used Cdc48-depleted cells to detect and measure amounts of ubiquitinated peptides present in various ribosomal fractions (Figure 2 and Figure 4). However, this experimental system might not reflect steady-state levels of either ubiquitinated polypeptides present on various ribosomal species or the ligase that promotes their ubiquitination. Therefore, we then performed sucrose gradient analysis to measure levels of Ltn1^FLAG^ associated with 60S and 80S in wild-type cells and in cells depleted of Ssb/RAC components, i.e., in *ssb∆∆*, *ssz1∆*, and *zuo1∆* strains. As in the previous experiment, we extracted and analyzed proteins from critical gradient fractions (Figure 5A–D and Figure A3). NIR signals corresponding to Ltn1^FLAG^ and Rpl3 present in 60S and 80S ribosomal fractions were expressed as Ltn1^FLAG^/Rpl3 ratios (Figure 5E). We did not probe for polyubiquitinated RNCs in this experiment, as the presence of functional Cdc48 in tested strains quickly removes ubiquitinated polypeptides from ribosomal species [13], making them undetectable by the conventional western blotting (please see Figure 1A in [50]). As with Cdc48-depleted cells, absence of Ssb/RAC in wild-type background reduced Ltn1^FLAG^ amounts in 80S fractions (Figure 4C and Figure 5E, compare bars from 80S). We also observed an altered ratio of the 60S:80S peak in *ssz1Δ* strain (Figure 5B) which could be reflective of some unique Ssz1’s function(s) outside of Ssb/RAC complex [57,58,59]. Nevertheless, the observed decrease was not specific for any particular Ssb/RAC deletion strain, suggesting that the recruitment of the ligase to the ribosome depends on a functional activity of Ssb/RAC triad.

Strikingly, this analysis also detected significant reduction in Ltn1^FLAG^ levels in fractions corresponding to the 60S ribosomal subunits upon deleting Ssb/RAC-encoding genes (Figure 5E, compare 60S bars). These data suggest that Ssb/RAC controls Ltn1 presence on both substrates of RQC, the 60S•RNCs and the 80S•RNCs complexes. The analysis of these data is presented in the Discussion.

## 3. Discussion

In this work we uncovered the relationships between ribosome-associated chaperone Ssb/RAC and Ub ligase Ltn1, the major executor of the RQC. We made several previously undescribed observations. We demonstrated that besides the polypeptides that remain associated with the free ribosomal subunit 60S, 80S-bound polypeptides also appeared to be the substrates for Ltn1, and their modification by this Ub ligase was accelerated by the ribosome-associated chaperone triad Ssb/RAC (Figure 2, Figure 3 and Figure 4). We also showed that loss of Ltn1-dependent ubiquitination observed in cells with dysfunctional Ssb/RAC is due to decreased amount of the ligase associating with 80S (Figure 4 and Figure 5) and 60S (Figure 5). Taken together, these data suggest that Ssb/RAC facilitates binding of Ltn1 to ribosomes. However, the suggested mechanism is not essential for Ltn1’s function on RNCs, as overexpressing Ltn1 in *ssb∆∆* cells rescues the growth suppression phenotype under proteotoxic stress conditions (Figure 1C–E). Furthermore, Ltn1^FLAG^ was detected on 60S and 80S ribosomal species in Ssb/RAC-null cells, although in smaller amounts than in wild-type cells (Figure 4 and Figure 5). These results imply that lack of functional Ssb/RAC complex does not abolish binding and catalytical activity of Ltn1 on a ribosome.

Thus, based on the combined data presented in this study, we propose that Ssb/RAC constitutes an *optional co-factor* that operates on ribosomes during RQC by detecting aberrant RNCs and increasing the efficiency of Ltn1 binding and ubiquitination of client polypeptides (Figure 6).

### 3.1. Non-Conventional 80S-RQC Pathway

Detection of ubiquitinated polypeptides associated with 80S and modified by Ltn1 (Figure 2, Figure 4 and Figure 5) suggests the existence of a pathway distinct from canonical RQC, as it operates on fully assembled ribosomes. We term this pathway 80S-RQC. Using a combination of genetic and biochemical approaches, we showed that similarly to canonical 60S-dependent RQC, 80S-RQC also uses the Ub ligase Ltn1 (Figure 2), whereby the ribosome-associated chaperone triad Ssb/RAC promotes recruitment of Ltn1 to 80S (Figure 4 and Figure 5). We propose that the selectivity of this pathway is based on high processivity and short interaction time of Ssb with normal/native RNCs (estimated average interaction time of Ssb with normal RNCs in yeast is 1–2 s [40]), while in case of nascent chain aberrations, the longer existing Ssb•RNC complex will allow accommodation of Ltn1 followed by RNC’s ubiquitination and degradation.

First described nearly a decade ago [4,5,60], RQC has been extensively studied in yeast and mammalian cells, as well as in cell-free settings. However, the general focus of the scientific community has focused on deciphering the mechanistic details of RQC that operates on the 60S subunit. In fact, biochemical studies conducted in multiple laboratories worldwide have detected association of Ltn1-ubiquitinated polypeptides predominantly with free 60S subunits [4,5,6], suggesting that 60S-RQC constitutes the major branch of the protein quality control on a ribosome. Biochemical cell-based studies were further supported by the exciting cryo-EM data that provided several images for 60S•peptidyl-tRNA•Rqc2 complexes [11,61] and Ltn1 Ub ligase [52,62]. However, the existence of a pool of ubiquitinated RNCs bound to fully assembled ribosome 80S were also observed for both model protein substrates [4,6,54] and endogenous proteins [50]. Present in significantly lower amounts than its 60S counterparts, 80S-associated Ub-modified RNCs were described as products of rejoining 60S•Ub-peptidyl-tRNA with the free 40S subunit, a phenomenon that may occasionally occur due to failure of timely interaction between 60S•Ub-peptidyl-tRNA and Rqc2, which prohibits subunits re-association based on cryo-EM data [52,55]. Therefore, splitting of ribosomal subunits was thought to function as a major RQC-specificity mechanism ensuring that 80S ribosomes translating normal polypeptides are not targeted by the surveillance system [3]. Given that Dom34-Hbs1 is unable to hydrolyze peptidyl-tRNA and thus generate 60S•Ub-peptidyl-tRNA species, whereby tRNA serves as Rqc2 interacting module, lack of functional Dom34-Hbs1 must abolish, or at least decrease, ubiquitination levels of RNCs present on 60S and 80S. This mechanistical concept has been shifted over the years as RQC research progressed, assigning stalling or prolonged ribosomal pausing as the RQC-specificity tool [3,63,64]. Consistent with this new RQC-specificity concept, studies conducted in our laboratory detected presence of endogenous Ltn1-ubiquitinated cellular polypeptides on 80S in yeast strains depleted of *DOM34*, *HBS1*, or both [50]. These data suggest that if Dom34-Hbs1-dependent splitting of stalled ribosomes is a prerequisite for Ltn1-dependent ubiquitination (Figure 1A), 80S•Ub-peptidyl-tRNAs cannot be formed via the 60S•Ub-peptidyl-tRNA•40S re-association mechanism in *dom34∆*, *hbs1∆*, or *dom34∆ hbs1∆* cells. In addition, recently published proximity-labeling proteomics, aimed to systematically characterize the core mammalian RQC complex, uncovered previously uncharacterized Ltn1-interacting proteins. Thus, over a dozen of r-proteins from the small ribosomal subunit 40S have been identified [49], further supporting our hypothesis that Ltn1 may interact with fully assembled ribosome 80S. Finally, Ltn1’ cryo-EM conducted in solution at 40 Å demonstrated the hinge-like structure of Ltn1 with extreme flexibility of both N-terminal (bordering 40S) and C-terminal (located near the exit tunnel) regions of the ligase [62], suggesting the possibility that this ligase might adapt multiple conformations depending on substrate geometry and context of the exit tunnel surrounding. All these findings support existence of an additional mechanism that likely represents a branch of the conventional RQC and operates on the 80S•RNC complexes by employing Ltn1. The findings described here identified Ssb/RAC as a co-factor of 80S•RNC mechanism.

### 3.2. Hsp70-Ssb and Ltn1 Binding to Ribosomes

Two recent studies have provided a detailed picture of Ssb interaction with ribosomes [38,44]. These studies found that Ssb makes several direct contacts with r-proteins and rRNAs, including Rpl35, Rpl39, and Rpl19, as well as expansion segments ES24 and ES41, all of which are located in close proximity to the exit tunnel of 60S [44]. Importantly, the dynamic nature of Ssb on a ribosome was demonstrated; it undergoes structural re-arrangement upon ATP hydrolysis. A combination of crystallography, cross-link studies, and computational biology revealed two conformations for Ssb. The *open conformation* predicts that Ssb is in the pre-hydrolysis state bound to ATP and its positively charged SBDα domain (Figure 1B) is oriented towards ES41 and is located between Rpl22 and Rpl31 [38,44]. In the *closed conformation*, Ssb is in the post-hydrolysis state bound to ADP, in which SBDα shifts by >60 Å, flipping onto SBDβ that is docked close to the exit tunnel. The closed conformation allows Ssb to interact with RNCs with high affinity. Conversion from the open to the closed conformation occurs upon ATP hydrolysis and is strictly RAC-dependent [36,44].

Detailed architecture of Ssb/RAC binding to yeast ribosome resembles characteristics of Ltn1 ligase binding to the 60S subunit. In fact, cryo-EM conducted with Ltn1 complexes purified from cells showed rRNA and r-proteins that interact with the C-terminus of Ltn1 near the exit tunnel, including Rpl22, Rpl31, Rpl3, Rpl24, and ES41 [52], whereby some sites are Ssb/RAC interactors. The Ltn1 cryo-EM results thus provoke an important question on how Ssb and Ltn1 co-exist on a ribosome, as they bind to the overall same region on 60S. Theoretically, two possibilities exist: (i) Ltn1 and Ssb/RAC work in synergy, both binding the ribosome; or (ii) they compete for the binding sites near the exit tunnel. Our data, which demonstrate functional cooperation between Ssb/RAC and Ltn1 (Figure 3, Figure 4 and Figure 5), strongly argue against the second model. Therefore, based on the model published in [44], we hypothesize that Ssb, in concert with RAC, performs the first steps of RQC; upon detecting aberrations within nascent chain, Ssb remains in its closed conformation. This conformational rearrangements within Ssb might have two alternative outcomes: (1) Ssb•ADP becomes suitable for direct binding of Ltn1, or (2) conversion of Ssb•ATP to Ssb•ADP opens regions near the ribosome exit tunnel for Ltn1 to bind. We were not able to detect a direct physical interaction between Ltn1 and Ssb1 either in vitro or in cells using ATP- or ADP-supplemented buffers. It will be important in the future to decipher molecular details of Ssb/RAC and Ltn1 interaction with ribosome using the proximity labeling approach and determine whether Ssb/RAC-mediated accommodation of Ltn1 on a ribosome depends on different conformations of Hsp70-Ssb, i.e., Ssb•ATP and Ssb•ADP.

### 3.3. Ssb/RAC Activity Is Associated with Free 60S Subunit

To detect polyubiquitinated RNCs associated with different ribosomal species, we used Cdc48-depleted cells as an experimental platform (Figure 2, Figure 3 and Figure 4). The Ub-selective chaperone Cdc48 promotes peptide extraction from the ribosome exit tunnel [13]. Thus, lack of functional Cdc48 leads to accumulation of Ub-modified peptides on ribosomal particles, which can be separated by centrifugation through sucrose gradients, and presence of Ub-RNCs in gradient fractions can be visualized by western blot analysis. Previously developed in our laboratory [50], this approach proved to be informative. However, this methodology may not reflect steady-state levels of proteins associated with ribosomes, such as Ltn1. To address the question of Ssb/RAC involvement in Ltn1 association with 60S and 80S in a physiologically relevant system, we measured Ltn1 levels in a wild-type strain in comparison to strains carrying deletion of *SSB1* and *SSB2* genes, *SSZ1*, and *ZUO1*. We found that Ssb/RAC mutations notably decreased the levels of Ltn1 associated not only with 80S ribosomes, but also with the free 60S subunit (Figure 5), while upon depletion of *CDC48*, amounts of 60S-bound Ltn1 were not significantly affected by RAC mutation (Figure 4). The observed discrepancy between the two strain backgrounds used (wild-type and *P_TET-07_-CDC48)* implies that lack of functional Cdc48 indeed affects steady-state levels of Ltn1 associated with ribosomal species, especially of 60S, which becomes saturated quickly. Considering that Ssb/RAC binds and operates on fully assembled ribosomes [25,36,40,41,42], the most straightforward interpretation of our data is that the chaperone’s primarily substrate is 80S•RNC, which, upon splitting by Dom34-Hbs1-Rli1-mediated activity, generates 60S•peptidyl-tRNA. In wild-type cells, tRNA present on this complex interacts with Rqc2, which functions as a co-factor for efficient Ltn1 recognition, recruitment and catalytic activity (Figure 1A). Lower levels of Ltn1 detected in Ssb/RAC-null strains (Figure 4) suggest that the presence of Ssb/RAC on a ribosome likely promotes Rqc2 binding. Future experimentation is needed to confirm this model.

## 4. Materials and Methods

### 4.1. Antibodies and Chemicals

In this study, we used the following antibodies: anti-ubiquitin (clone P4G1, Biolegend, San-Diego, CA, USA, Cat# 646301); anti-FLAG (Sigma, St Louis, MO, USA, Cat# F7425); anti-RPL3 (ScRPL3, Iowa, USA, Developmental Studies Hybridoma Bank, University of Iowa); anti-rabbit IgG-HRP (GE-Healthcare, Chicago, IL, USA, Cat# NA934V); anti-mouse IgG-HRP (GE-Healthcare, Cat# NA931); anti-rabbit IR-Dye 800CW, (LI-COR, Lincoln, NE, USA, Cat# 925-32211); anti-mouse IR-Dye 680RD, (LI-COR, Cat# 925-68070).

Sodium Chloride (Beantown Chemical, Hudson, NE, USA, Cat# 214380) was used at a final concentration of 0.8 M; Hygromycin B (ThermoFisher, Waltham, MA, USA, Cat# 10687010) was used at a final concentration of 25 or 100 µg/mL; Doxycycline (Acros Organics, Radnor, PA, USA, Cat# 446060050) was used at a final concentration of 10 µg/mL; Cycloheximide (Sigma, Cat# 01810-5G) was used at a final concentration of 100 µg/mL; RiboLock RNase inhibitor (Thermo Fisher, Cat#EO0381) and SIGMAFAST™ Protease Inhibitor Cocktail Tablets, EDTA-Free (Sigma, Cat# S8830) were used at concentrations recommended by the manufacturers.

### 4.2. Plasmids

To generate FLAG-tagged constructs of Ltn1, we used the reverse (3′) primer containing the sequence that encoded the FLAG tag. To clone Ltn1-FLAG into pUAD (URA) plasmid [65], we placed Xho1 site into forward primer and FLAG sequence followed by the Kpn1 site into the reverse primer. PCR products were digested with Xho1 and Kpn1 and cloned into Xho1 and Kpn1 sites of the pUAD plasmid. To clone Ltn1-FLAG into pNS1-LEU plasmid, we incorporated Xho1 site into forward primer and FLAG sequence followed by the Nhe1 site into reverse primer. PCR products were digested with Xho1 and Nhe1 and cloned into Xho1 and Nhe1 sites of the pNS1-LEU plasmid. pNS1-LEU plasmid was generated by cloning *ADH1* promoter into PESC-LEU (Stratagene, San-Diego, CA, USA) instead of GAL1-GAL10 promoters using Not1 and Xho1 sites.

### 4.3. Yeast Strains, Medium, and Growth Conditions

We used standard recipes for YPD (1% yeast extract, 2% peptone, 2% dextrose) and synthetic glucose media. Cells were grown at 30 °C unless mentioned otherwise. All strains used in this study are listed in Table 1.

To generate deletion strains, *his5+* or *LEU2* disruption cassettes were integrated via homologous recombination at the targeted genomic loci using standard PCR-based techniques [66]. Strains were verified using PCR-based approaches as described in [50].

Where indicated, yeasts were transformed with pUAD, PUAD-Ltn1^FLAG^, pNS1 or pNS1-Ltn1^FLAG^ plasmids using standard protocol [67], and transformed cells were selected by their ability to grow on minimal media plates lacking uracil or leucine, respectively.

For doxycycline depletion of Cdc48, *P_TET-O7_-CDC48* cells and their derivative strains were grown at 30 °C overnight, diluted to OD_600_ ~0.05 and grown overnight in the presence of doxycycline. Cells were diluted to OD_600_ ~0.5 in YPD medium supplemented with doxycycline, grown for an additional 4 h at 30 °C and taken through the further analysis.

### 4.4. Cell Viability and Growth Assays

For cell viability assays, yeast strains were grown in YPD or minimal selective media overnight at 30 °C shaking as noted in Figure Legends. Overnight cultures were grown to saturation, then backdiluted with fresh YPD media to OD_600_ ~0.3 and grown for an additional 3.5 h at 30 °C. The cultures were adjusted to OD_600_ ~0.5 prior further manipulations. Six 1:5 dilutions were prepared and spotted onto respective agar plates. Plates were incubated at 20 °C or 30 °C temperature for 2–3 days and scanned.

For the growth assay in liquid cultures, cells were grown similarly to the growth assay described above, except that all yeast cultures tested were diluted to final OD_600_ ~0.2. NaCl or HygB were added to some of the cultures’ aliquots, as indicated in Figure Legends and 200 µL of each culture were placed into 96-well plate in triplicates. The cultures were grown in BioTek Synergy HT microplate reader (Winoosky, VT, USA) at 30 °C with shaking. OD_600_ were recorded automatically every 5 min for 24 h. Lag time and time at V_max_ were determined using the instrument’s Gen5 software. Recorded OD_600_ during the log phase was used to derive the doubling time using the formula:doubling time = [{duration × log (2)}/{log (final OD_600_) − log (initial OD_600_)}](1)

### 4.5. Sucrose Gradient Centrifugation

Sucrose gradient centrifugation analysis was done as described before [50]. Briefly, cell cultures were harvested after treatment with cycloheximide (100 µg/mL) for 5 min. Cells were lysed by bead beating in lysis buffer containing 100 mM NaCl, 3 mM MgCl_2_, 10 mM Tris–HCl pH = 7.4, 0.2 mg/mL of heparin and 100 µg/mL of cycloheximide. Clarified whole-cell extracts equivalent to 150 µg of RNA was loaded onto 15–35% or 15–42% sucrose gradients prepared in 70 mM NH_4_Cl, 4 mM MgCl_2_ and 10 mM Tris–HCl (pH 7.4) buffer. Gradients were centrifuged at 36,000 rpm for 4 h 15 min using the Beckman SW41Ti rotor at 4 °C. Gradients were fractionated using a Beckman fraction recovery system connected to an EM-1 UV monitor (Bio-Rad, Hercules, CA, USA), fractions were collected as indicated in Figure Legends and processed for further analyses.

### 4.6. Protein Extraction and Analysis

To prepare whole-cell protein lysate, yeast cells were collected, resuspended in cold buffer A (50 mM Tris–HCl, pH = 8.0; 150 mM NaCl; 3 mM MgCl_2_ and SIGMAFAST™ Protease Inhibitor Cocktail) and lysed by glass-beads shearing. Lysates were clarified by centrifugation and total protein concentrations were measured by Bradford assay. Ten mg of total protein lysate were separated by SDS-PAGE. Proteins were detected by western blotting using antibodies indicated in Figure Legends.

To analyze protein content in sucrose gradient fractions, we precipitated proteins from individual fractions with 10% TCA in the presence of 5 μg of BSA carrier. Proteins were pelleted by centrifugation at 15,000 rpm for 20 min at 4 °C using Eppendorf 5424 centrifuge. Pellets were washed once with ethanol and once with acetone, air-dried at 37 °C and resuspended in 1xSDS-PAGE loading dye. Proteins were resolved in polyacrylamide gels under denaturing conditions, transferred onto nitrocellulose membrane (Whatman, Maidstone, United Kingdom), blocked with 10% milk for 30 min, and incubated with primary antibodies overnight at 4 °C. Membranes were washed, incubated with appropriate secondary antibodies. For conventional non-quantitative western blotting we used HRP-conjugated secondary antibodies. Signals were visualized by ECL (Millipore, Burlington, MA, USA). For quantitative western blot analysis, we used either anti-rabbit or anti-mouse secondary antibodies conjugated to IR-Dyes 800CW or 680RD, respectively. Signals were visualized using Typhoon imager (GE Healthcare) by scanning blots at 800 nm or 680 nm, respectively.

## Figures and Tables

**Figure 1 ijms-21-06815-f001:**
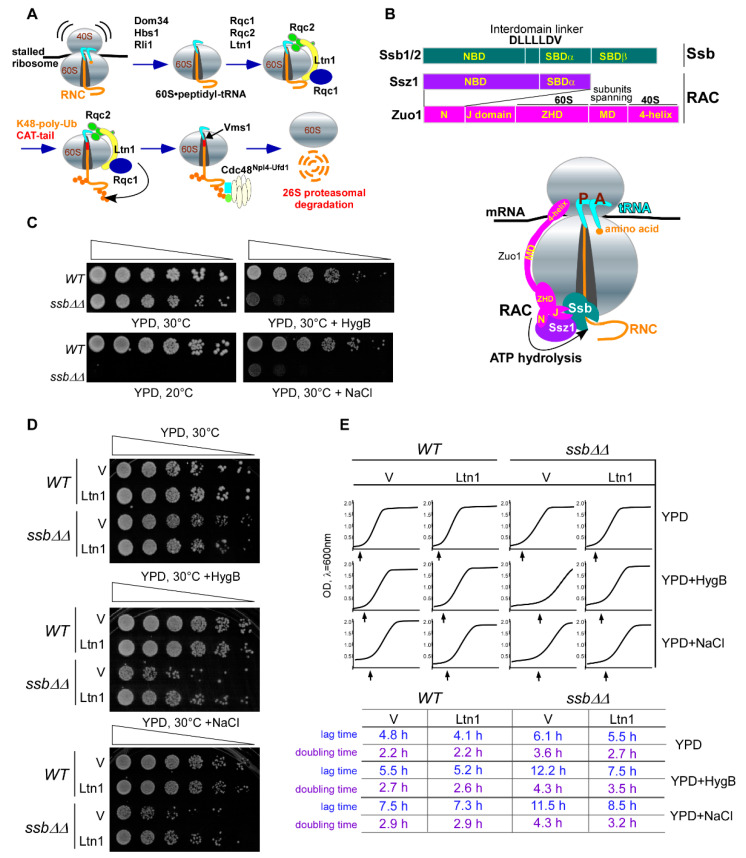
Schematic representation of (**A**) RQC (details are in the text), and (**B**) the functional domains of Ssb/ribosome-associated complex (RAC) and interaction of Ssb/RAC with the 80S ribosome. Ssb1 is in green, Ssz1 is in purple, and Zuo1 is in pink. Functional domains are marked in yellow. Cartoon at the bottom depicts positioning of the Ssb/RAC complex on the 80S ribosome. (**A**,**B**) tRNAs located at A and P sites of the ribosome are shown in turquoise; ribosome-associated nascent chain (RNC) is shown in orange. Overexpression of Ltn1 in *ssb∆∆*cells rescues stress-induced growth suppression. (**C**) Wild-type (BY4741) and *ssb1∆ ssb2∆* (*ssb∆∆*) yeast strains were grown in YPD. Cell cultures were adjusted to the same cell density, and five-fold dilutions were spotted onto YPD agar plates and on YPD plates supplemented with either 25 μg/mL of hygromycin B (HygB) or 0.8 M NaCl. Plates were incubated at 30 °C or at 20 °C for 48 h, as indicated. (**D**) Wild-type and *ssb∆∆* cells were transformed with empty vector control (V) or Ltn1^FLAG^-expressing constructs. Cells were grown in SC ura^-^ medium and adjusted to the same cell density. Serial dilutions were spotted onto plates as described in (**B**). (**E**) Cell cultures from (**C**) were adjusted to OD_600nm_ ~0.2 and grown in liquid cultures in YPD, YPD + HygB (25 μg/mL), or YPD + NaCl (0.8 M) medium for 24 h with continued shaking at 30 °C. OD_600nm_ were measured every 5 min. The representative growth curves are shown on the top, arrows indicate the end of lag phase; lag and doubling time parameters are shown at the bottom.

**Figure 2 ijms-21-06815-f002:**
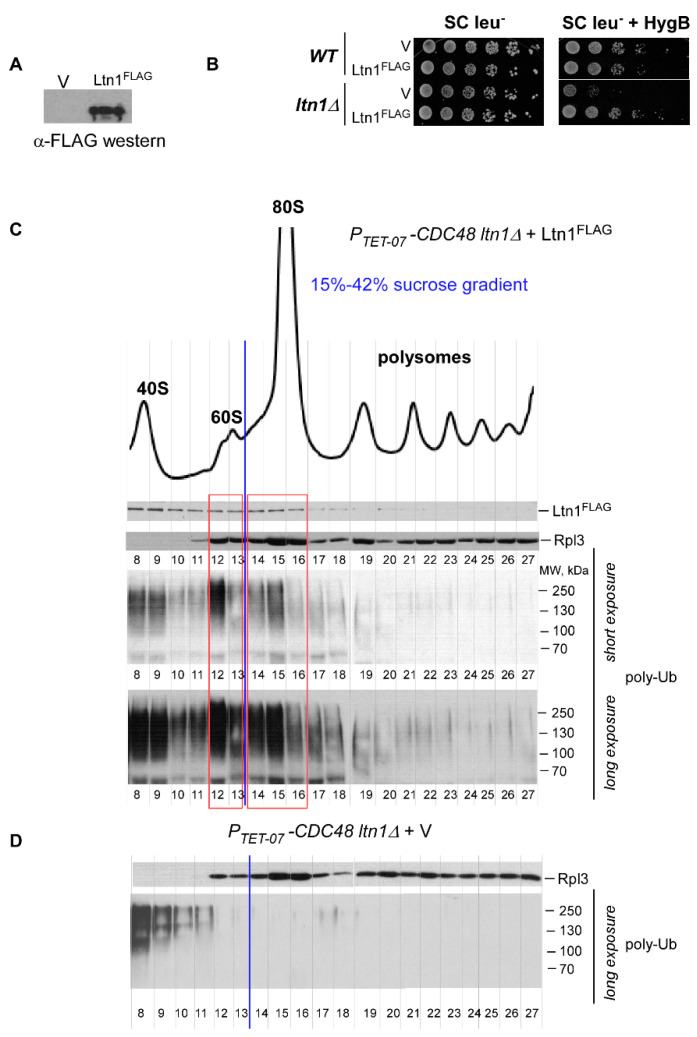
Ectopically expressed Ltn1^FLAG^ is functionally active: it rescues hygromycin-induced lethality in *ltn1∆* cells and ubiquitinates polypeptides associated with 60S and 80S ribosomal species. (**A**) Ltn1^FLAG^ cloned under control of the constitutive *ADH* promoter in 2 μc plasmid was expressed in *P_TET-07_-CDC48* cells grown in SC leu^-^ medium. Expression was detected by western blotting using anti-FLAG primary and HRP-fused secondary antibodies. Empty vector (V) was used as a negative control. (**B**) *ltn1∆* cells and their parental strain BY4741 (wild-type, *WT*) were transformed with empty vector control (V) or with 2 μc plasmid constitutively expressing Ltn1^FLAG^. Transformants were grown in SC leu^-^ medium and adjusted to the same cell density. Five-fold dilutions were spotted onto SC leu^−^ control agar plates and onto SC leu^−^ plates supplemented with 100 μg/mL HygB. Plates were grown at 30 °C for 3 days. (**C**,**D**) *P_TET-0_-CDC48 ltn1∆* cells transformed with Ltn1^FLAG^-expressing construct (**C**) or empty vector control (**D**) were grown in SC leu^-^ medium in the presence of Dox for 16 h, diluted to OD_600nm_~0.3, and grown for 4 h in YPD+Dox. Before harvesting, cells were treated with cycloheximide (CHX) for 5 min. Whole cellular lysates were centrifuged through a 15–42% sucrose gradient (10.8 mL) and fractionated. Total protein extracts were isolated from 23 individual fractions (fractions 5–27; ~390 μL each). Proteins were separated on 10% SDS-polyacrylamide gels in duplicate and transferred onto nitrocellulose membranes. One membrane was probed with anti-FLAG antibodies to detect Ltn1^FLAG^ (top panel). The bottom part of the second membrane was probed with anti-Rpl3 antibodies (middle panel), while the top part of the same membrane was probed with anti-Ub antibodies (bottom panels; two exposures are shown). We used HRP-fused secondary antibodies and ECL system to visualize protein signals. The experiment was repeated 3 times; representative images are shown. Peak levels of ribosomal species detected by absorbance measurements at 254 nm are shown at the top of the figure. The blue vertical line indicates visual separation between 40S–60S and 80S-polysome fractions of the gradient, while 60S- and 80S-containing fractions are boxed in red.

**Figure 3 ijms-21-06815-f003:**
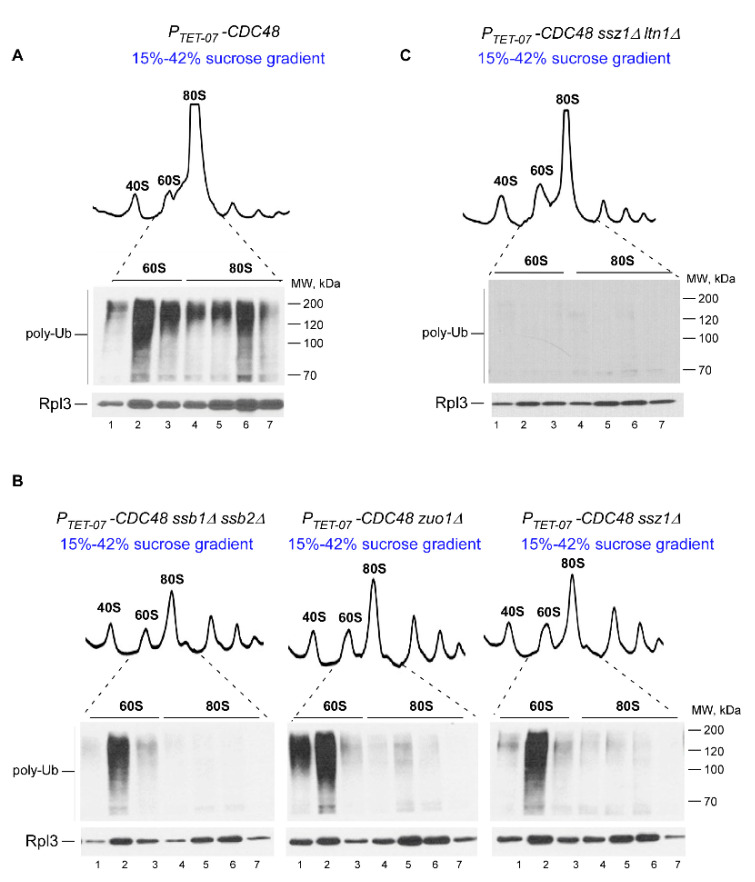
Lack of Ssb/RAC components significantly decreases ubiquitination of polypeptides associated with the 80S monosome. Sucrose gradient sedimentation analysis of ribosomes extracted from: (**A**) *P_TET-07_-CDC48* cells and their derivative strains containing additional deletions of (**B**) *SSB1* and *SSB2*, or *ZUO1*, or *SSZ1*, or (**C**) *SSZ1* and *LTN1*. Cells were grown in YPD media in the presence of Dox for 16 h, diluted to OD_600nm_~0.3, and grown for 4 h in YPD+Dox. Lysates were centrifuged through 15–42% sucrose gradients (10.8 mL) and fractionated into 14 fractions (~780 μL). Total protein was isolated from each fraction and analyzed as described in Figure 2C,D. Western blotting images of 7 critical fractions containing 60S (1–3) and 80S (4–7) ribosomal species are shown. The experiment was repeated 3 separate times; representative images are shown.

**Figure 4 ijms-21-06815-f004:**
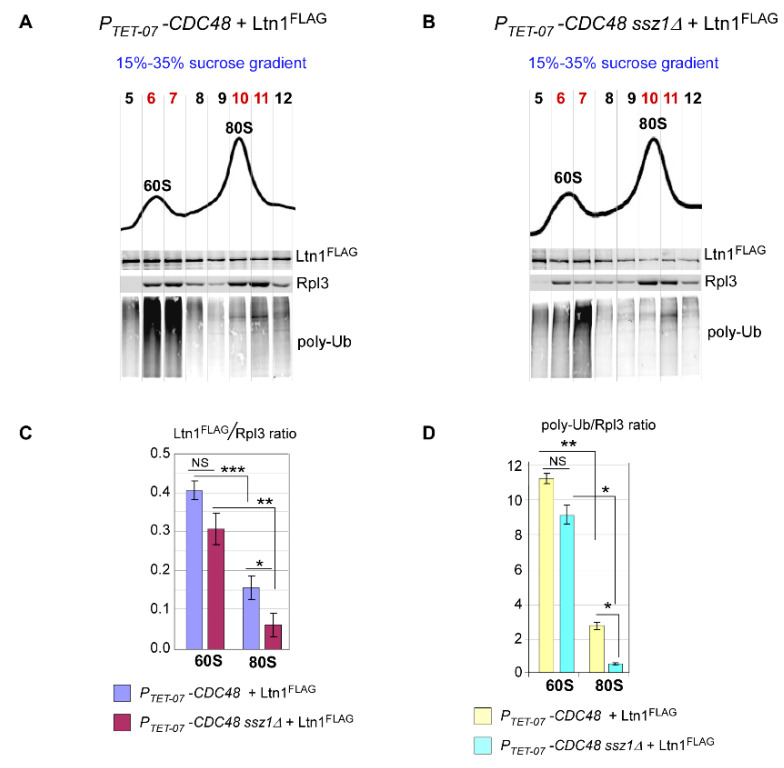
Lack of Ssz1 co-chaperone of the Ssb/RAC complex results in reduced levels of polyubiquitinated nascent chains on 80S ribosomes due to decreased levels of Ltn1^FLAG^. (**A**,**B**) *P_TET-07_ -CDC48* and *P_TET-07_-CDC48 ssz∆* cells transformed with Ltn1^FLAG^-expressing construct were grown in SC leu^-^ medium in the presence of Dox for 16 h, diluted to OD_600nm_ ~0.3, and grown for additional 4 h in YPD+Dox. Whole-cell lysates were separated by centrifugation through 15–35% sucrose gradients and fractionated with continuous measurement of absorbance at 254 nm to visualize ribosomal peaks. Fractions containing 60S–80S ribosomal species (5–12) were analyzed further. Proteins extracted from these fractions were separated on SDS-polyacrylamide gels in duplicate and analyzed by quantitative western blotting. One membrane was probed with anti-FLAG antibodies (top panels), while the top part of the second membrane was probed with anti-Ub antibodies and the bottom part with anti-Rpl3 antibodies (middle and bottom panels). We used IR-Dye-680RD secondary antibodies for Rpl3- and Ub-probed blots, and IR-Dye-800CW antibodies for FLAG-probed blots. IR signals were detected on Typhoon imager by scanning membranes at 800 nm and 680 nm; ImageQuant was used to analyze the images. Experiments were repeated 3 independent times; representative images are shown. (**C**,**D**) Bands corresponding to Ltn1^FLAG^, Rpl3, and poly-Ub species were converted to phosphorimager units. To normalize 60S-associated Ltn1^FLAG^ to Rpl3 levels, we used phosphorimager units derived from Ltn1 and Rpl3 signals present in fractions 6 and 7. To normalize 80S-associated Ltn1^FLAG^ to Rpl3, we used units derived from Ltn1 or Rpl3 signals present in fractions 10 and 11. (**C**) Ltn1^FLAG^/Rpl3 ratios were plotted as bar graphs. (**D**) Phosphorimager units derived from poly-Ub signals present in fractions 6 + 7 and 10 + 11 of each gradient (panels A and B) were plotted as bar graphs. *p* values: ***, <0.001; **, <0.01; *, <0.05; NS: not significant. Two-tailed two-sample unequal variance *t*-test was used for statistical analysis. Error bars represent SD.

**Figure 5 ijms-21-06815-f005:**
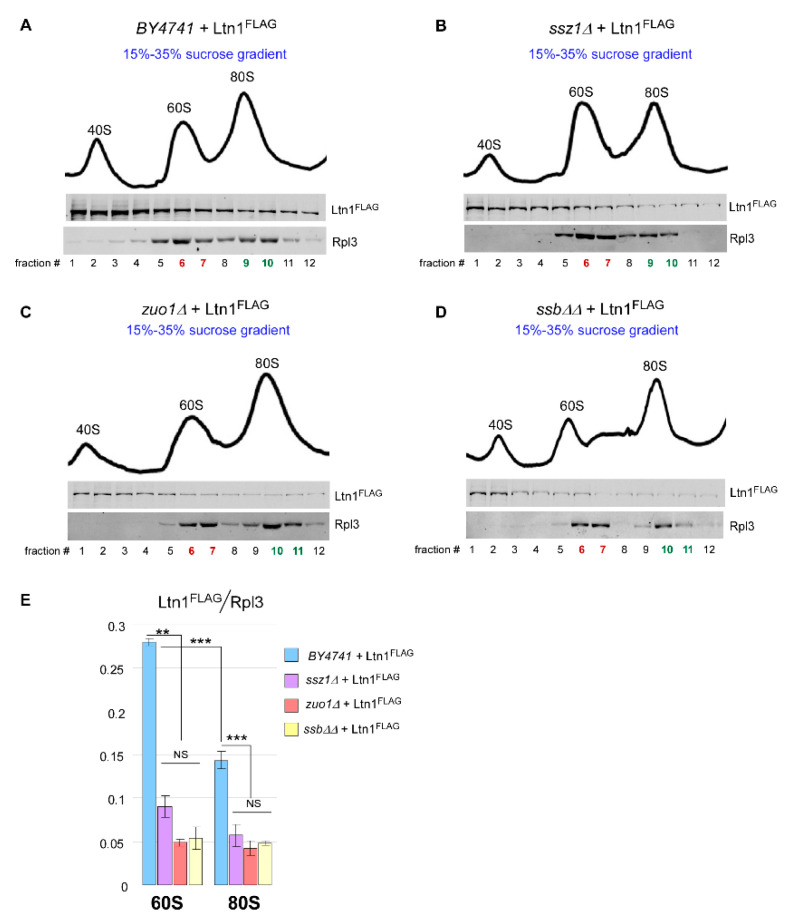
Deleting genes encoding Ssb/RAC leads to decreased association of Ltn1^FLAG^ with 60S and 80S ribosomal species. (**A**–**D**) Indicated yeast strains transformed with Ltn1^FLAG^-expressing construct were analyzed as described in Figure 4. Experiment was repeated 3 independent times; representative images are shown. (**E**) Bands corresponding to Ltn1^FLAG^ and Rpl3 were converted to phosphorimager units. Signal averages were calculated for Ltn1^FLAG^ and Rpl3 present in fractions highlighted in red (for 60S) and green (for 80S). Ltn1^FLAG^ levels were then normalized to Rpl3 in the same fractions and expressed as Ltn1^FLAG^/Rpl3 ratios plotted as bar graphs. *p* values: ***, <0.001; **, <0.01; NS: not significant. Two-Tailed two-sample unequal variance *t*-test was used for statistical analysis. Error bars represent SD.

**Figure 6 ijms-21-06815-f006:**
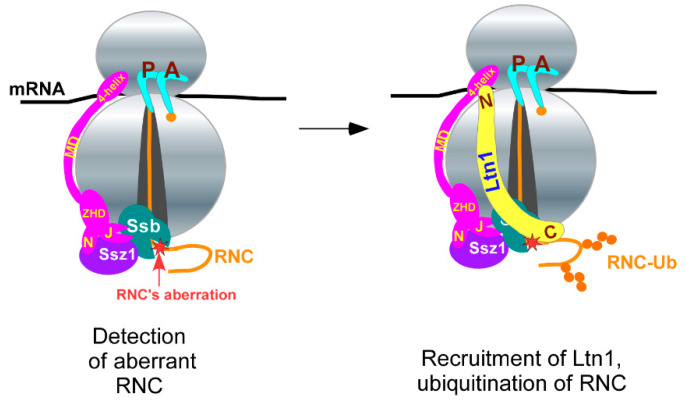
Model for cooperative function of Ssb/RAC and Ltn1 Ub ligase. See details in the text.

**Table 1 ijms-21-06815-t001:** Yeast strains used in this study.

Strain	Genotype	Reference
BY4741	*MAT*a *his3-1 leu2-0 met15-0 ura3-0*	Thermo Fisher
*zuo1Δ*	*MAT*a *his3-1 leu2-0 met15-0 ura3-0 zuo1Δ::KANMX6*	Thermo Fisher
*ssz1Δ*	*MAT*a *his3-1 leu2-0 met15-0 ura3-0 ssz1Δ::KANMX6*	Thermo Fisher
*ssb1Δ ssb2Δ*	*MAT*a *his3-1 leu2-0 met15-0 ura3-0 ssb1Δ::KANMX6 ssb2Δ::LEU2*	This study
*P_TET-O7_-CDC48*	*MAT*a *URA3::CMV-tTA his3-1 leu2-0 met15-0 pCDC48::KanR-tet07-TATA*	Thermo Fisher
*P_TET-O7_-CDC48 ltn1Δ*	*MAT*a *URA3::CMV-tTA his3-1 leu2-0 met15-0 pCDC48::KanR-tet07-TATA ltn1Δ::his5+*	This study
*P_TET-O7_-CDC48 ssb1Δ ssb2Δ*	*MAT*a *URA3::CMV-tTA his3-1 leu2-0 met15-0 pCDC48::KanR-tet07-TATA ssb1Δ::his5+ ssb2Δ::LEU2*	This study
*P_TET-O7_-CDC48 ssz1Δ*	*MAT*a *URA3::CMV-tTA his3-1 leu2-0 met15-0 pCDC48::KanR-tet07-TATA ssz1Δ::his5+*	This study
*P_TET-O7_-CDC48 zuo1Δ*	*MAT*a *URA3::CMV-tTA his3-1 leu2-0 met15-0 pCDC48::KanR-tet07-TATA zuo1Δ::LEU2*	This study
*P_TET-O7_-CDC48 ssz1Δ ltn1Δ*	*MAT*a *URA3::CMV-tTA his3-1 leu2-0 met15-0 pCDC48::KanR-tet07-TATA ssz1Δ::his5+ ltn1Δ::LEU2*	This study

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
