# Peer review of "Cooperativity between the Ribosome-Associated Chaperone Ssb/RAC and the Ubiquitin Ligase Ltn1 in Ubiquitination of Nascent Polypeptides"

_ijms, 2020, doi:10.3390/ijms21186815_

Round 1

Reviewer 1 Report

The authors show that lack of the Ssb1 and Ssb2 chaperones inhibits growth under stress conditions (high NaCl, presence of hygromycin B, cold stress). Overexpression of the Ltn1 ubiquitin ligase, a major component of the ribosome-associated protein quality control system (RQC), can partially alleviate the growth defect of ssbDD cells under stress. They provide evidence that Ltn1 can associate not only with 60S subunits but also with fully assembled 80S ribosomes. They show that Ltn1-dependent polyubiquitination of RNCs on 80S ribosomes is enhanced in the presence of the Ssb/RAC complex and provide data suggesting that this is because Ssb/RAC promotes the recruitment of Ltn1 to 80S ribosomes. They also show that association of Ltn1 with 60S ribosomes is reduced in cells lacking Ssb/RAC complex components when Cdc48 is present but less so when it is absent.

The work will be of interest to scientists working on nascent polypeptide quality control and translation.

Questions and suggestions:

Figure 1: Does Lnt1 overexpression corrects the cold sensitivity of ssbDD cells ?

Figure 2: Some explanations in the text as to what is the function of Cdc48 would be useful. The gradient profile shows a deficit of free 60S subunits relative to free 40S subunits (and a “shoulder effect” on the 60S peak; the same phenomenon is seen in Figure A1). Is that due to Cdc48 depletion (but this phenomenon is not seen on Figure 3A)? The authors should comment and explain. Figure 2D: why was the peak tracing (254 nm absorbance) of the control strain transformed with the empty vector not shown ?

Figure 3: The authors should provide a brief conclusion at the end of the associated paragraph (page 7) regarding the finding that lack of Ssb/RAC complex components specifically prevents accumulation of polyubiquitinated polypeptides associated with 80S ribosomes.

Figure 5: Why were levels of poly-UBs not analysed ? is that because their levels are too low? Figure 5B: the 254 nm absorbance profile shows a very strong 60S peak relative to the 80S peak in ssz1D cells, not seen when other Ssb/RAC complex components are absent. Can the authors comment?

Page 10, line 321: The sentence “However, this experimental system….ubiquitination” is clumsy and should be rephrased.

Dicussion section:

Lines 351-352: “ and their modification is mediated by the ribosome-associated chaperone triad Ssb/RAC”: this statement should be altered as it seems to suggest that Ssb/RAC provides the catalytic activity for ubiquitination.

Author Response

Reviewer 1.

The authors show that lack of the Ssb1 and Ssb2 chaperones inhibits growth under stress conditions (high NaCl, presence of hygromycin B, cold stress). Overexpression of the Ltn1 ubiquitin ligase, a major component of the ribosome-associated protein quality control system (RQC), can partially alleviate the growth defect of ssbDD cells under stress. They provide evidence that Ltn1 can associate not only with 60S subunits but also with fully assembled 80S ribosomes. They show that Ltn1-dependent polyubiquitination of RNCs on 80S ribosomes is enhanced in the presence of the Ssb/RAC complex and provide data suggesting that this is because Ssb/RAC promotes the recruitment of Ltn1 to 80S ribosomes. They also show that association of Ltn1 with 60S ribosomes is reduced in cells lacking Ssb/RAC complex components when Cdc48 is present but less so when it is absent. The work will be of interest to scientists working on nascent polypeptide quality control and translation.

We thank the Reviewer for the insightful evaluation of the manuscript. We are grateful for the suggestions on how to improve this work.

  1. Figure 1: Does Lnt1 overexpression corrects the cold sensitivity of ssbDD cells?

Answer: Based on our observations, overexpression of Ltn1FLAG did not correct the cold sensitivity of ssbDD. We interpreted these results as that Hsp70-Ssb participates in more than one pathway responsible for adaptation to cold (as described in the recent review, PMID: 17298585), wherein a shortage in RQC constitutes only a part of the observed phenotype. To keep the Results-2.1 section focused, we decided not to modify the text stating this observation.

  1. Figure 2: Some explanations in the text as to what is the function of Cdc48 would be useful.

Answer: We extended the background information on Cdc48 and the Ub-binding adaptors Npl4-Ufd1 in the Introduction section of the revised version of the manuscript, which reads like this: “A member of the AAA+ family of molecular chaperones [14], Cdc48/p97 forms hexamer and upon ATP hydrolysis generates a large amount of energy, underlying its activity in energy-dependent processes [15]. The Ub-binding selectivity of Cdc48 is provided by Ub-binding co-factors, such as Npl4/NPL4 and Ufd1/UFD1 [16,17]”. Please, see page 2, lines 49-52, marked in red. In addition, we added a new panel in Figure 1 (panel A) that schematically illustrates the activities of the RQC members.  

The gradient profile shows a deficit of free 60S subunits relative to free 40S subunits (and a “shoulder effect” on the 60S peak; the same phenomenon is seen in Figure A1). Is that due to Cdc48 depletion (but this phenomenon is not seen in Figure 3A)? The authors should comment and explain.

Answer: The Reviewer is correct; we have detected a significantly larger peak for the 40S than for the 60S subunit in the experiments depicted in Figure 2C. Based on our extensive previous experience with this system, we are confident that the observed effect is not due to Cdc48 depletion. Our work with Cdc48-depleted cells (PTET-07-CDC48) presented here (Figure 3) and the previous work published in Nucleic Acid Research in 2016 (PMID:27325745) demonstrate that depletion of Cdc48 does not cause a deficit of free 60S subunits relative to free 40S subunits under the conditions of growth in the rich YPD medium. However, in the case of yeasts transformed with plasmids, we maintained cultures in an appropriate minimal SD medium. This, in our opinion, is the main cause of the discrepancy between sucrose gradients and peak appearance. In fact, in bacteria, minimal medium conditions resulted in alterations of ribosome biogenesis dynamics (PMID:17804668). In Figures 2C and A1, we used Cdc48-depleted strain carrying an additional deletion of LTN1 (PTET-07-CDC48 ltn1D). The following sentence was added to the main text: “We also noticed significantly larger peak for the 40S subunit when compared with peak for the 60S subunit (Figures 2C and A1) that was not caused by Cdc48 depletion [51]. Considering that in this experimental setting we maintained yeast culture transformants in the SC medium for a prolonged time, we concluded that minimal nutritional supplements might result in alterations of ribosome biogenesis dynamics, similarly to bacterial cells [53]; (page 5, lines 192-197, marked in red).   

Figure 2D: why was the peak tracing (254 nm absorbance) of the control strain transformed with the empty vector not shown?

Answer: Due to a shortage of space in Figure 2, the peak tracing of the control strain transformed with empty vector control is presented in the Supplementary Figure A1. For clarity to a reader, we mentioned it in the main text of the manuscript: “Of note, the peak levels of ribosomal species detected by absorbance measurements at 254 nm in the control strain transformed with empty vector are presented in Figure A1”; (page 5, lines 190-192, marked in red).

  1. Figure 3: The authors should provide a brief conclusion at the end of the associated paragraph (page 7) regarding the finding that lack of Ssb/RAC complex components specifically prevents accumulation of polyubiquitinated polypeptides associated with 80S ribosomes.

Answer: We are thankful to the Reviewer for this great suggestion. We added the concluding sentence at the end of the Results-2.3 section that reads like this: “Taken together, these data demonstrate that lack of Ssb/RAC complex components specifically prevents the accumulation of polyubiquitinated polypeptides associated with 80S ribosomes”. The new sentence is on page 7, lines 256-258, marked in red.

  1. Figure 5: Why were levels of poly-UBs not analysed? is that because their levels are too low?

Answer: The Reviewer is correct, the presence of functional Cdc48 in wild-type cells quickly removes ubiquitinated polypeptides from ribosomal species, making them undetectable by the conventional western blotting. To address this concern, we added a new sentence to clarify this issue: “ We did not probe for polyubiquitinated RNCs in this experiment, as the presence of functional Cdc48 in tested strains quickly removes ubiquitinated polypeptides from ribosomal species [13], making them undetectable by the conventional western blotting (please see Figure 1A in [51])”; (page 10, lines 343-346, marked in red).

Figure 5B: the 254 nm absorbance profile shows a very strong 60S peak relative to the 80S peak in ssz1D cells, not seen when other Ssb/RAC complex components are absent. Can the authors comment?

Answer: We thank the Reviewer for bringing up this important point.  Previous works by other groups (PMID:22203981, PMID:20038635, PMID:18833196) have shown that, in addition to their joint function as part of the Ssb/RAC triad, Ssz1, Hsp70-Ssb, and Zuo1 also have independent roles in controlling various cellular processes. Therefore, we suggest that the altered 60S:80S ratio unique to ssz1D could be reflective of the Ssz1 function beyond RQC. We would be excited to follow up on this observation in the future. We have added the following sentence in the Results-2.5 section: “We also observed an altered ratio of the 60S:80S peak in ssz1Δ strain (Figure 5B) which could be reflective of some unique Ssz1’s function(s) outside of Ssb/RAC complex [57-59]”; (page 10, lines 347-349)​.

  1. Page 10, line 321: The sentence “However, this experimental system….ubiquitination” is clumsy and should be rephrased.

Answer: For the Reviewer’s recommendation, the sentence was rephrased to provide a better rationale of the experiments presented in the Results-2.5 section, and now reads like this: “However, this experimental system might not reflect steady-state levels of either ubiquitinated polypeptides present on various ribosomal species or the ligase that promotes their ubiquitination”; (page 10, lines 336-338, marked in red).

  1. Dicussion section: Lines 351-352: “ and their modification is mediated by the ribosome-associated chaperone triad Ssb/RAC”: this statement should be altered as it seems to suggest that Ssb/RAC provides the catalytic activity for ubiquitination.

Answer: We are grateful to the Reviewer for picking up upon such an unfortunate choice of words. The new sentence reads like this: “We demonstrated that besides the polypeptides that remain associated with the free ribosomal subunit 60S, 80S-bound polypeptides also appeared to be the substrates for Ltn1, and their modification by this Ub-ligase was accelerated by the ribosome-associated chaperone triad Ssb/RAC (Figures 2, 3, and 4)”; (page 11, lines 370-373, marked in red).

Reviewer 2 Report

Molecular mechanisms for quality control of newly synthesized proteins has been under investigation for a decade or longer. There are several components for this surveillance mechanism, some of which are associated with the actively translating ribosomes. Two such mechanisms involve the Ssb/RAC complex and the Ltn1 ubiquitin ligase, both of which can detect misfolded nascent peptides. The subunits of ribosomes carrying nascent proteins tagged with ubiquitin by these mechanisms are dissociated by Dom34:Hbs1 followed by elimination of the ubiquitinated peptidyl-tRNA and proteasome degradation of the peptide. However, the interaction between the RAC and Ltn1 processes two mechanisms has not been fully investigated.

In the current manuscript Ghosh and Shcherbik present evidence for cooperativity between the RAC and Lnt mechanisms. Specifically, they have used yeast to show that overexpression of Lnt1 rescues ssb mutants. Using a CDC48 mutant to decrease the degradation of ubiquitinated peptides, they also quantified ubiquitinated peptides on ribosomes in Ltn1 and ssb/zuo/ssc mutants and showed cooperativity between the binding of Ltn1 and RAC complexes to 60S and 80S ribosomes. I find that, together, these well-conceived and well-executed experiments document the conclusions. However, I have some minor points, listed below, that should be addressed prior to publication. I also want to mention that I appreciate that the authors have used both the yeast and they human nomenclature throughout the manuscript which helps compare their results with other published experiments on human cells and cell free extracts.

  • A schematic presentation of the relevant surveillance pathways would help follow the arguments and experiment design.
  • The white/grey lettering in the hot pink and other colored backgrounds in Fig 1 are not very legible. More contrast, e.g. using yellow letters would help a lot.
  • The 60S peak, particularly in Fig 2 has some clear shoulders and the 40S peak is significantly larger than the 60S peak. Is this reproducible? If so, it might indicate ribosomal assembly problems or 60S instability. Please comment.
  • Define where the lab time ends in Fig 1D, i.e. how is the lag time defined functionally
  • Are all cultures grown to the same overnight density? If some cultures have entered stationary phase, the lag time will be affected by parameters other than the mutations used.
  • Line 168: Naturally? Means in wildtype background?
  • Line 180: 60S and 40S.
  • Line 190: The Ltn1/Ub ratio is approximately the same in 40S and 60S. Why is the 40S left out for the remainder of the manuscript?
  • Line 203: typo lnt>ltn
  • Line 233: we deleted SSB1 and SSB2 on …..> we deleted SSB1 and SSB2 in …..
  • Line 251: mL>µL

Author Response

Reviewer 2.

Molecular mechanisms for quality control of newly synthesized proteins has been under investigation for a decade or longer. There are several components for this surveillance mechanism, some of which are associated with the actively translating ribosomes. Two such mechanisms involve the Ssb/RAC complex and the Ltn1 ubiquitin ligase, both of which can detect misfolded nascent peptides. The subunits of ribosomes carrying nascent proteins tagged with ubiquitin by these mechanisms are dissociated by Dom34:Hbs1 followed by elimination of the ubiquitinated peptidyl-tRNA and proteasome degradation of the peptide. However, the interaction between the RAC and Ltn1 processes two mechanisms has not been fully investigated.

In the current manuscript Ghosh and Shcherbik present evidence for cooperativity between the RAC and Lnt mechanisms. Specifically, they have used yeast to show that overexpression of Lnt1 rescues ssb mutants. Using a CDC48 mutant to decrease the degradation of ubiquitinated peptides, they also quantified ubiquitinated peptides on ribosomes in Ltn1 and ssb/zuo/ssc mutants and showed cooperativity between the binding of Ltn1 and RAC complexes to 60S and 80S ribosomes. I find that, together, these well-conceived and well-executed experiments document the conclusions. However, I have some minor points, listed below, that should be addressed prior to publication. I also want to mention that I appreciate that the authors have used both the yeast and they human nomenclature throughout the manuscript which helps compare their results with other published experiments on human cells and cell free extracts

Thank you for finding our work well-thought and well-executed.

  1. A schematic presentation of the relevant surveillance pathways would help follow the arguments and experiment design.

Answer: Thank you for this insightful suggestion. We added a new panel that illustrates the major steps of the RQC. The new panel A is now placed in Figure 1.

  1. The white/grey lettering in the hot pink and other colored backgrounds in Fig 1 are not very legible. More contrast, e.g. using yellow letters would help a lot.

Answer: We agree with the Reviewer that letter colors in the former Figure 1/panel A (current Figure 1/panel B) are awkward and lack of contrast. We modified this diagram to address the Reviewer’s concern.

  1. The 60S peak, particularly in Fig 2 has some clear shoulders and the 40S peak is significantly larger than the 60S peak. Is this reproducible? If so, it might indicate ribosomal assembly problems or 60S instability. Please comment.

Answer: The Reviewer is correct we have detected significantly larger peaks for 40S than for 60S subunits. The same concern was also raised by the Reviwer#1. However, we should bring the Reviewer’s attention that in our previous experiments (Figure 3 of the current manuscript, and Figures in PMID:27325745 published by us in 2016) yeast cultures were maintained in the rich YPD medium. However, in the case of yeasts transformed with plasmids (Figure 2C), we maintained cultures in an appropriate minimal SD medium. This, in our opinion, is the main cause of the discrepancy between sucrose gradients and peak appearance. Likely, minimal medium conditions resulted in alterations of ribosome biogenesis dynamics, a phenomenon that was observed in bacterial cells (PMID:17804668) and is a subject for further investigation in yeast. The following sentence was added to the main text: “We also noticed significantly larger peak for the 40S subunit when compared with peak for the 60S subunit (Figures 2C and A1) that was not caused by Cdc48 depletion [51]. Considering that in this experimental setting we maintained yeast culture transformants in the SC medium for a prolonged time, we concluded that minimal nutritional supplements might result in alterations of ribosome biogenesis dynamics, similarly to bacterial cells [53]; (page 5, lines 192-197, marked in red).

  1. Define where the lag time ends in Fig 1D, i.e. how is the lag time defined functionally

Answer: For the Reviewer’s recommendation, we marked the end of the lag time in Figure 1E (former Figure 1D).

Are all cultures grown to the same overnight density? If some cultures have entered stationary phase, the lag time will be affected by parameters other than the mutations used.

Answer: Yes, all cultures tested were grown overnight to saturation. To reflect this experimental detail, we modified the sentence in the Materials and Methods section 4.4. “Cell viability and growth assays”. The modified text reads like this: “Overnight cultures were grown to saturation, then back-diluted with fresh YPD media to OD600 ~0.3 and grown for an additional 3.5 hrs at 30°C. The cultures were adjusted to OD600 ~0.5 prior further manipulations”; (page 15, lines 531-533, marked in red).

  1. Line 168: Naturally? Means in wildtype background?

Answer: The Reviewer is correct; this sentence is confusing. By “Naturally”, we meant endogenous polypeptides (not overexpressed protein-reporters) that undergo modification with ubiquitin. We rephrase this sentence to make it clear to a reader: “This assay is based on separating total cellular yeast lysate into ribosomal species-specific fractions and analyzing ubiquitination status of endogenous RNCs generated in cells [51]”; (page 5, lines 171-173; marked in red).

  1. Line 180: 60S and 40S

Answer: To address this concern of the Reviewer, we re-structured the text describing the results depicted in Figure 2C. We placed the sentence on the presence of a 40S-polyubiquitin signal after describing the detection of Ub-modified RNCs in 60S fractions. We believe that this textual rearrangement will help a reader in results interpretation. The section reporting our new finding of 80S-associated ubiquitinated nascent chains constitutes a new paragraph. Modified text (pages 5-6, lines 183-199) is marked in red.

  1. Line 190: The Ltn1/Ub ratio is approximately the same in 40S and 60S. Why is the 40S left out for the remainder of the manuscript?

Answer: We decided not to follow ubiquitin signal co-sedimented with 40S fractions because Figures 2C and A1 demonstrate that this signal is not a product of Ltn1 Ub-ligase. We visualized poly-Ub species in fractions 8-11, which correspond to 40S, in the absence of Ltn1, i.e. in PTET-07-CDC48 ltn1D strain transformed with empty vector control (see Figures 2C and A1, bottom panels).

  1. Line 203: typo lnt>ltn

Answer: Corrected, marked in red (page 6, line 215);

  1. Line 233: we deleted SSB1 and SSB2 on …..> we deleted SSB1 and SSB2 in …..

Answer: Corrected, marked in red (page 7, line 245);

  1. Line 251: mL>µL

Answer: Corrected, marked in red (page 8, line 265).

We are very grateful to the Reviewer for picking up these typos.

Reviewer 3 Report

This is very interesting manuscript describing new data on co-translation protein folding control. The manuscript is well-written, and all data presented exhaustively. Experimental design is well thought out, all conclusions made by authors are supported by results. Nevertheless, I have some questions and minor corrections.

  1. Lane 123 - Figure 1 legend looks a bit broken by bold type before (B) panel.
  2. On Figure 1D - it's not easy to compare 12 curves. Maybe, curves can be grouped, e.g. V & Ltn1, and colourized.
  3.  Figure 2 C&D - significant amount of Ltn1-FLAG was detected in 40S fractions. Is Ltn1 associated with free 40S subunits? Moreover, poly-Ub signals were detected in both strains' 40S fractions, and these signals look comparable. How authors could explain it?
  4. Lane 536 - misprint: "Branford" instead of "Bradford". Conventional Bradford method is not compatible with detergents concentration specified in buffer composition.

Overall, presented manuscript could be accepted with minor corrections.

Author Response

Reviewer 3.

This is very interesting manuscript describing new data on co-translation protein folding control. The manuscript is well-written, and all data presented exhaustively. Experimental design is well thought out, all conclusions made by authors are supported by results. Nevertheless, I have some questions and minor corrections.

Thank you for your kind comments.

  1. Lane 123 - Figure 1 legend looks a bit broken by bold type before (B) panel.

Answer: For the Reviewer's suggestion, we reformatted the Figure 1 legend (page 4, lines 127-128).

  1. In Figure 1D - it's not easy to compare 12 curves. Maybe, curves can be grouped, e.g. V & Ltn1, and colourized.

Answer: Thank you for this suggestion. We spent a good amount of time formatting this Figure by grouping curves in various ways and by using color code. We concluded that, although “bulky” at the first glance, the present format illustrates the data most straightforwardly. Besides, below the growth curves image, we also provide the growth characteristics of tested cultures (i.e. lag time and doubling time) in the table format, which matches the curve panel to ease the comparison. 

  1. Figure 2 C&D - significant amount of Ltn1-FLAG was detected in 40S fractions. Is Ltn1 associated with free 40S subunits?

Answer: Based on the current literature describing Ltn1/Listerin interaction with ribosomes (PMID:23319619 and PMID:25349383), it is unlikely that the ligase directly binds the 40S subunit. However, although intriguing, this possibility might exist, and it is up for further investigations. To keep the manuscript focused, we decided not to follow this lead.

Moreover, poly-Ub signals were detected in both strains' 40S fractions, and these signals look comparable. How authors could explain it?

Answer: This question was also raised by Reviewer 2 (comment 7). We explain these results by the fact that 40S-associated polyubiquitinated species are not the products of Ltn1, as they can be detected in fractions 8-11 that correspond to the 40S subunit (see Figure 2C and A1, bottom panels) in the absence of Ltn1, i.e. in PTET-07-CDC48 ltn1D strain transformed with empty vector control.

  1. Lane 536 - misprint: "Branford" instead of "Bradford". Conventional Bradford method is not compatible with detergents concentration specified in buffer composition.

Answer: We are very grateful to the Reviewer for identifying this unfortunate error in the method description. We checked our records and realized that for total protein extract preparation cells were lyzed in buffer A that is detergent-free (Figure 2A). We modified section 4.6. “Protein extraction and analysis” accordingly (page 16, lines 555-556). The total protein concentration was indeed measured by Bradford (not Branford!) assay. The typo was also corrected (page 16, line 558). Thank you again for detecting these mistakes.